# TrueGradeAI: Retrieval-Augmented and Bias-Resistant AI for Transparent and Explainable Digital Assessments

## Abstract

This paper introduces TrueGradeAI, an AI-driven digital examination frame- work that directly addresses the shortcomings of traditional paper assessments, namely excessive paper usage, logistical complexity, grading delays, and eval- uator bias. The system preserves natural handwriting by capturing stylus input on secure tablets and applying transformer-based optical character recognition for transcrip- tion. Evaluation is performed through a retrieval-augmented pipeline that inte- grates faculty solutions, cache layers, and external references, enabling a large language model to assign scores with explicit, evidence-linked reasoning. In con- trast to prior tablet-based exam systems that primarily digitize responses, True- GradeAI advances the field by incorporating explainable automation, bias mitiga- tion, and auditable grading trails. By combining handwriting preservation with scalable and transparent evaluation, the framework reduces environmental costs, accelerates feedback cycles, and progressively builds a reusable knowledge base, while explicitly working to mitigate grading bias and ensure fairness in assess- ment.

## 1 Introduction

### 1.1 The Limitations of Paper-Based Assessment

Despite advances in digital infrastructure, most institutions continue to rely on paper-based exami- nations as their primary mode of assessment. This dependency creates three persistent challenges. First, the environmental burden is substantial: large-scale exam sessions require massive volumes of paper, and studies show that more than 30% of printed pages in answer booklets remain un- used (Sharma & Patil, 2017). Beyond raw consumption, the life cycle of paper from production to landfill generates emissions and waste that are increasingly unsustainable. Second, logistics are cumbersome. Institutions must organize printing, secure transport, and large-scale storage, creating administrative overhead that scales poorly with student populations. Third, manual grading is slow and inconsistent. It often takes weeks before students receive feedback, and human evaluators in- troduce subjective bias, with systematic disparities documented across gender and socioeconomic lines (Nguyen & Clarke, 2021). These factors together highlight the urgent need for fairer, faster, and greener alternatives.

### 1.2 TrueGradeAI: A Transparent Digital Framework

To address these shortcomings, we introduce TrueGradeAI, a digital examination framework de- signed to preserve the authenticity of handwritten responses while enabling transparent and auditable evaluation. As illustrated in Figure 1, students use stylus-enabled tablets to replicate the natural act of writing, and their responses are digitized using state-of-the-art handwriting recognition models (Li et al., 2021a). Once transcribed, answers are passed through a retrieval-augmented pipeline that anchors evaluation in faculty-prepared solutions, institutional knowledge bases, and trusted refer- ences. A large language model then assigns scores and produces concise rationales that explicitly link outcomes to retrieved evidence. Unlike existing tablet-based systems that merely digitize re- sponses, TrueGradeAI emphasizes explainability, bias mitigation, and auditability. Every decision

is logged for potential review, ensuring accountability while reducing grading latency from weeks to hours. The result is an end-to-end platform that eliminates the inefficiencies of paper while building trust through transparent evaluation.

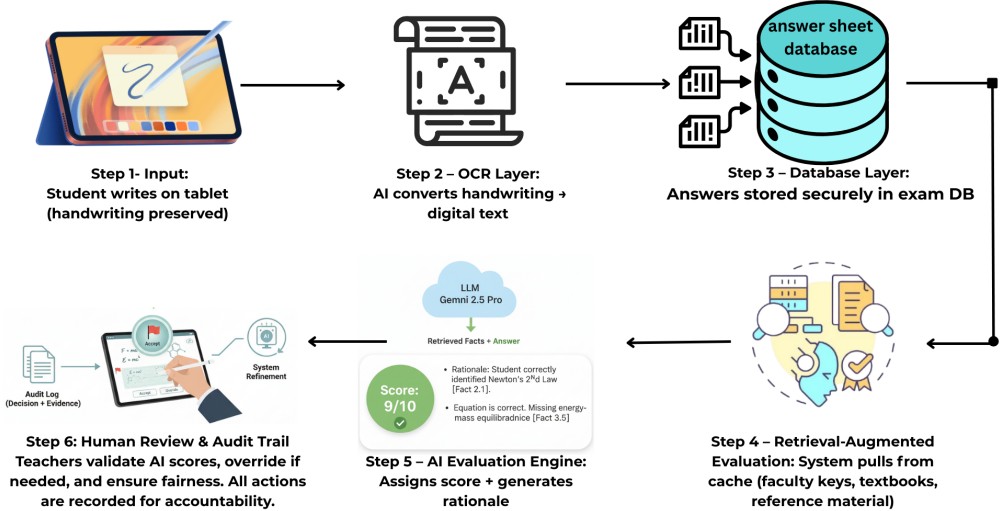

Figure 1: The TrueGradeAI pipeline

## 1.3 KEY CONTRIBUTIONS

The contributions of this paper are threefold:

- A complete digital assessment pipeline: TrueGradeAI manages secure exam delivery, handwriting capture, transcription, automated grading, and reporting without reliance on paper.
- Explainable and reliable scoring: The system leverages retrieval-augmented generation (Lewis et al., 2020) to generate fact-linked rationales, addressing opacity in existing AI-based grading tools.
- Bias-aware and auditable design: Calibration routines, decision logs, and human-in-the-loop verification collectively reduce grading bias (Nguyen & Clarke, 2021) and maintain accountability.

## 2 RELATED WORK

### 2.1 DIGITAL AND AUTOMATED ASSESSMENT SYSTEMS

Automated grading systems have evolved over several decades, initially focusing on objective question types. With the advent of large language models, systems now provide more nuanced evaluation and explanations (Lewis et al., 2020). Platforms such as ExamX (Gupta & Kumar, 2023) and LMExams, which has been piloted at MAHE (Rao & of Higher Education, 2022), offer paperless exam delivery and stylus capture. However, they typically lack retrieval-augmented grading and explicit bias mitigation. In contrast, TrueGradeAI integrates handwriting recognition (Li et al., 2021a) with retrieval-augmented scoring and evidence-linked rationales, establishing a transparent, auditable pipeline for evaluation.

### 2.2 ADVANCES IN HANDWRITING RECOGNITION

Handwriting recognition is a critical component of digital assessment systems. Early Handwritten Text Recognition (HTR) methods combined convolutional neural networks (CNNs) with recurrent

neural networks (RNNs), achieving good performance on controlled datasets but struggling with diverse handwriting styles and multilingual scripts common in academic contexts.

Recent transformer-based approaches have improved robustness significantly. Microsoft's TrOCR (Li et al., 2021a) leverages vision and text transformers in a unified framework, achieving state-of-the-art accuracy on printed and handwritten datasets. TrueGradeAI adopts TrOCR as its OCR backbone and preserves both pen-stroke data and page images. This ensures robust transcription today and enables future multimodal extensions that combine visual, temporal, and linguistic features.

### 2.3 EXPLAINABLE GRADING AND RETRIEVAL-AUGMENTED SYSTEMS

The field of automated grading has developed considerably, beginning with feature-based Automated Essay Scoring (AES) systems that extracted surface features such as word counts, syntactic complexity, and discourse structure (Page, 1966; Shermis & Burstein, 2013). While these approaches provided some efficiency gains, their lack of interpretability and limited accuracy on open-ended responses raised concerns about fairness and trust (Zhang & Litman, 2020).

Recent advances in large language models (LLMs) have enabled systems that can produce not only scores but also explanatory rationales. However, many of these systems remain vulnerable to generating unsupported or inconsistent justifications, since they are not explicitly grounded in reference material.

Retrieval-Augmented Generation (RAG) addresses this gap by linking generative outputs to curated knowledge sources, thereby improving factual reliability (Lewis et al., 2020). Frameworks such as *COMMENTATOR* (Srivastava et al., 2024) have demonstrated how retrieval-assisted methods can support human annotators in dataset creation, but they are primarily designed for annotation tasks rather than evaluation.

In contrast, TrueGradeAI directly integrates RAG into the grading pipeline. Student responses are compared against faculty-prepared solutions, prior submissions, and reference materials stored in cache layers. The pipeline produces both a score and a fact-linked rationale, ensuring transparency and reducing evaluator bias. Combined with calibration workflows, audit logs, and appeals management, this design provides a more trustworthy and auditable alternative than existing solutions such as ExamX and LMExams.

Table 1: Comparison with existing digital examination platforms.

| Capability | ExamX | LMExams (MAHE) | TrueGradeAI |
|---|---|---|---|
| Stylus handwriting preserved | Partial | ✓ | ✓ |
| Handwriting → OCR quality | Limited OCR | Basic OCR | **Transformer OCR (TrOCR)** |
| Retrieval-augmented grading | ✗ | ✗ | ✓**(two-tier caches)** |
| Explainable reasoning | ✗ | ✗ | ✓**(fact-linked rationales)** |
| Bias-mitigation workflow | ✗ | ✗ | ✓**(calibration & audits)** |
| Audit trail & provenance | Logs only | Logs only | ✓**(decision & evidence logs)** |
| End-to-end latency | Days-weeks | Days-weeks | **Hours (batched)** |
| Environmental impact | Reduced paper | ✓(Paperless) | ✓**(Paperless + digital feedback)** |
| Teacher review tools | Limited | Limited | ✓**(confidence flags, appeals)** |
| Deployment model | Cloud SaaS | Campus devices | **Hybrid (cloud + on-prem)** |

## 3 TRUEGRADEAI PORTALS AND ARCHITECTURE

TrueGradeAI integrates a modular ecosystem of portals and backend services designed to streamline the digital assessment lifecycle. The system is organized around two key ends: the student portal and the teacher portal, connected through secure pipelines that guarantee fairness, transparency, and auditability.

### 3.1 STUDENT PORTAL: SECURE ENROLLMENT AND STYLUS-BASED EXAMINATION

The student-facing portal initiates the exam process with multi-factor biometric verification, such as facial recognition and ID card OCR, to ensure candidate authenticity (Abdelrahman & Lindqvist,

2017). Once authenticated, students enter a locked exam environment on institution-provided tablets equipped with a stylus-enabled handwriting canvas (Liu et al., 2020).

As illustrated in Figure 2, the interface allows candidates to attempt exams securely while maintaining a natural writing experience. The workflow includes:

- Authentication: Exam access is granted only after biometric checks and admit card validation.
- Handwriting capture: Students write answers using a stylus, preserving natural handwriting input.
- Digital transcription: Responses are converted into machine-readable text using Microsoft's TrOCR (Li et al., 2021b).
- Secure storage: Both handwriting strokes and OCR-transcribed text are encrypted and stored in a structured exam database for auditability.

### 3.2 Teacher Portal: Randomized Distribution and Review Dashboard

On the teacher's end, submitted scripts are anonymized and reshuffled before being distributed to reviewers, ensuring evaluator bias linked to student identity is minimized.

As shown in Figure 3, the teacher-facing dashboard integrates live monitoring, result management, student analytics, and AI-augmented grading results. Key functionalities include:

- Randomized allocation: Ensures that no reviewer consistently grades the same student.
- Result management: Provides a consolidated view of student submissions and progress.
- Student analytics: Displays aggregated performance trends, difficulty levels, and re-evaluation statistics.
- Appeals and oversight: Facilitates transparent re-evaluation workflows with evidence-linked audit trails.

Unlike traditional grading where teachers manually sift through scripts, TrueGradeAI enhances usability by combining AI-assisted scoring with human oversight, balancing efficiency and accountability.

### 3.3 Portal Functionalities at a Glance

To illustrate the distinct perspectives of students and teachers, Figures 2 and 3 provide visual snapshots of the two portals. The student portal emphasizes secure enrollment and handwriting capture, while the teacher portal highlights AI-augmented grading and oversight.

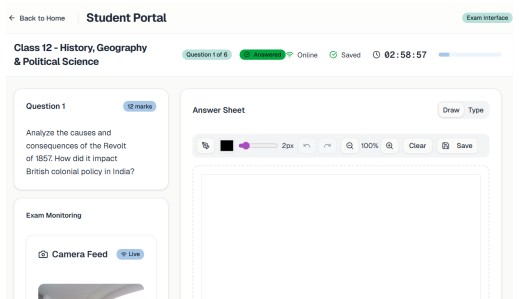

Figure 2: Student Portal: Stylus-based exam interface with biometric authentication, handwriting canvas, and live monitoring.

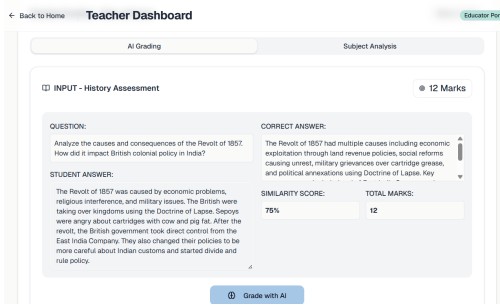

Figure 3: Teacher Portal: AI grading dashboard showing similarity scoring, reference answers, and analytics.

Table 2 further summarizes the functionalities of both portals, complementing the visual illustrations in Figures 2 and 3.

Table 2: Portal functionalities of TrueGradeAI at a glance

| Portal | Functionality | Purpose |
|---|---|---|
| Student Portal | Biometric authentication (face ID + ID OCR) | Ensure candidate authenticity |
| | Stylus-based handwriting input | Preserve natural writing experience |
| | TrOCR transcription of responses | Enable machine-readable grading |
| | Encrypted storage of scripts | Prevent tampering, ensure auditability |
| Teacher Portal | Randomized script allocation | Mitigate evaluator bias |
| | Result management dashboard | Track submissions and evaluation progress |
| | RAG-powered answer retrieval | Ground grading in verified sources |
| | LLM-based scoring with rationales | Provide explainable outcomes |
| | Appeals and oversight dashboard | Support transparency and accountability |

# 4 MODEL ARCHITECTURE

The modular architecture of TrueGradeAI is presented in Figure 4. The framework is designed as a multi-stage pipeline that integrates secure data capture, retrieval-augmented reasoning, and explainable grading. The workflow ensures that student inputs are authenticated, transcribed, semantically compared with authoritative answers, and evaluated with transparent rationales.

The architecture is described across three major components: *Initialization*, *Answer Processing and Retrieval*, and the *LLM Evaluation Module*.

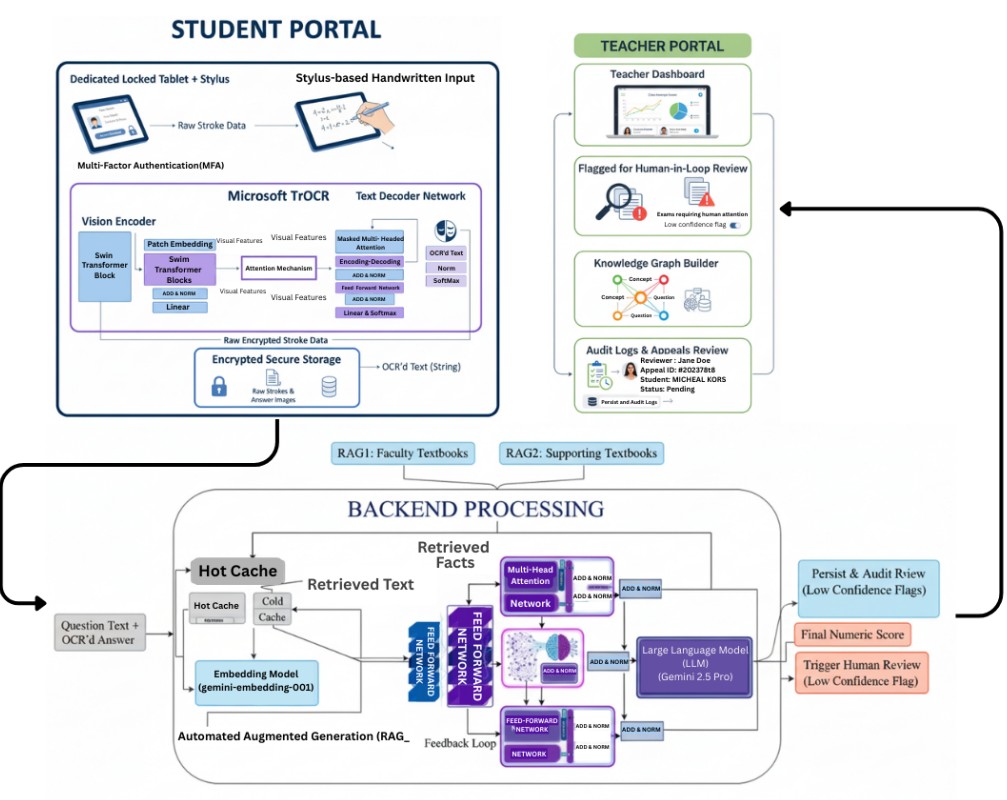

Figure 4: TrueGradeAI architecture: modular workflow integrating student portal, teacher portal, retrieval modules, and LLM-based evaluation.

## 4.1 INITIALIZATION

The system initializes by preparing two retrieval-augmented generation (RAG) modules:

- RAG1: Faculty Knowledge Base. Model answers curated by teachers are chunked into question-level units and indexed for direct retrieval.
- RAG2: Supporting Materials. Supplementary textbooks and references are chunked by topic and stored in a cache-augmented configuration. Each question initializes a *COLD cache* with top-$k$ relevant embeddings, while the *HOT cache* remains empty until promoted facts are frequently accessed.

This initialization ensures that every question begins with both authoritative solutions and broader supporting context, enabling low-latency and context-specific fact retrieval.

## 4.2 ANSWER PROCESSING AND FACT RETRIEVAL

Once initialization is complete, student responses are processed through the following steps:

**Vector Embedding –** Handwritten responses captured via stylus are first transcribed using TrOCR (Li et al., 2021b). The transcribed text is then encoded into dense vector representations using Google's `gemini-embedding-001` model (Google DeepMind, 2025), which provides semantically rich embeddings optimized for retrieval and alignment with reference answers. This combination ensures robust handling of handwriting variability while enabling efficient semantic comparison in downstream RAG modules.

**RAG1 Checking –** Each student response is first compared against RAG1 chunks. A similarity score determines alignment with faculty-prepared answers. If the threshold (e.g., 20%) is exceeded, the answer is marked correct and forwarded with its partial/full score to the LLM.

**HOT and COLD Cache –** For answers below threshold, the system checks COLD and HOT caches. Frequently accessed facts are promoted into HOT memory for faster retrieval, while broader context remains in COLD storage. This policy prevents cache bloating and ensures efficiency.

**Fallback Mechanism –** If neither RAG1 nor caches improve the score, RAG2 performs a deeper retrieval over supporting textbooks. Newly useful facts are added back into COLD memory for future queries, enriching the institutional knowledge base.

## 4.3 LLM EVALUATION MODULE

The final stage integrates all available information:

- Faculty-prepared solutions (RAG1).
- Student transcriptions.
- Retrieved supporting facts from HOT/COLD caches.
- Fallback retrieval from RAG2.

A large language model (LLM), such as Gemini 2.5 Pro (**?**), evaluates responses by generating:

1. A numerical score aligned with marking schemes.
2. A structured explanation that highlights correct reasoning, omissions, and areas for improvement.
3. An evidence-linked audit log for transparency.

## 5 DATASET AND EVALUATION

### 5.1 DATASET DESCRIPTION

We constructed a dataset of 10,000 question-answer (QA) pairs from NCERT Class 12 *History, Political Science, and Geography* textbooks (NCERT, 2021b;c;a). Each entry includes the original

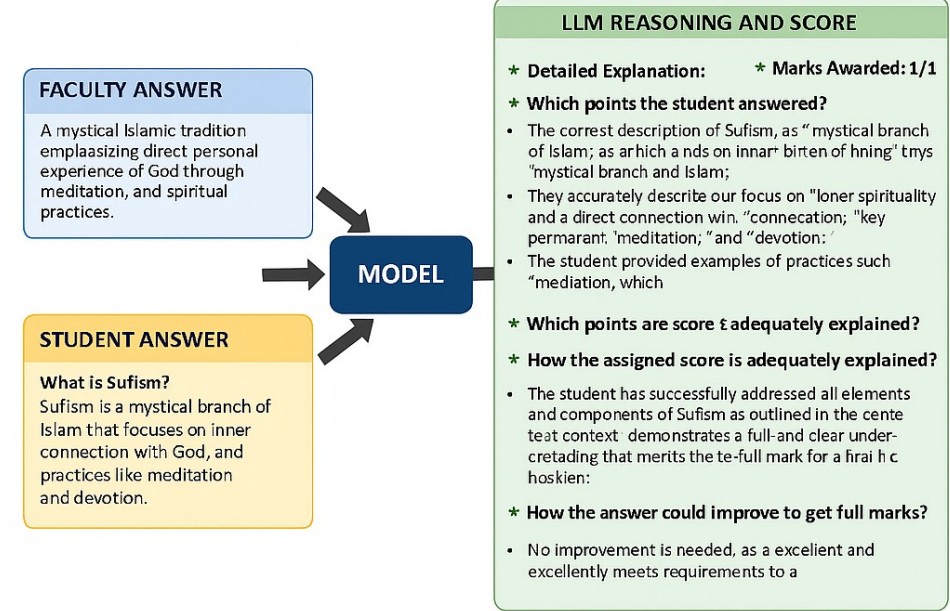

Figure 5: LLM reasoning module: comparison of student answer with faculty answer, followed by structured rationale and score generation.

question, a textbook-aligned reference answer, and a factual correctness score. To ensure consistency, the dataset was cleaned for noise, standardized in format, and assigned sequential identifiers (S. No.). Metadata at the category level (Fail, Average, Good, Excellent) supports structured benchmarking. The collection spans both factual and conceptual domains, ensuring strong curriculum alignment.

## 5.2 VALIDATION SETUP

To test dataset reliability, we compared AI-based semantic similarity scoring with human evaluation. Two independent teachers graded responses conservatively, penalizing brevity and insufficient elaboration. This setup simulates real-world faculty grading practices and allows us to measure alignment between automated scoring and stricter human judgment (Bailey & Meurers, 2008).

## 5.3 EVALUATION METRICS

We report agreement between AI and faculty evaluation using correlation and inter-rater metrics (Cohen, 1960; Ben-David, 2009):

Table 3: Agreement metrics between AI-based scoring and faculty evaluation.

| Metric | Value |
| --- | --- |
| Pearson correlation | 0.982 |
| Spearman correlation | 0.985 |
| Cohen's Kappa | 0.688 |

These results indicate strong consistency and substantial agreement, suggesting that the dataset is both factually reliable and resilient to grading bias.

## 5.4 RESULTS AND DISCUSSION

The validation experiments confirm strong alignment between AI-based scoring and faculty evaluation. The scatter plot in Figure 6 shows that most points fall close to the 45° line of perfect

agreement, demonstrating high linear consistency. The confusion matrix in Figure 7 highlights categorical agreement: while "Fail" cases were consistently identified by both AI and faculty, stricter human grading reduced the number of "Excellent" assignments (only 108 cases). Mid-range answers (Average/Good) were more harshly evaluated by faculty, reflecting realistic grading conservatism.

Together with the reported agreement metrics (Pearson = 0.982, Spearman = 0.985, Cohen's Kappa = 0.688), these results demonstrate that the dataset is both factually robust and resilient to human grading bias, making it suitable for training and evaluating explainable grading systems such as TrueGradeAI.

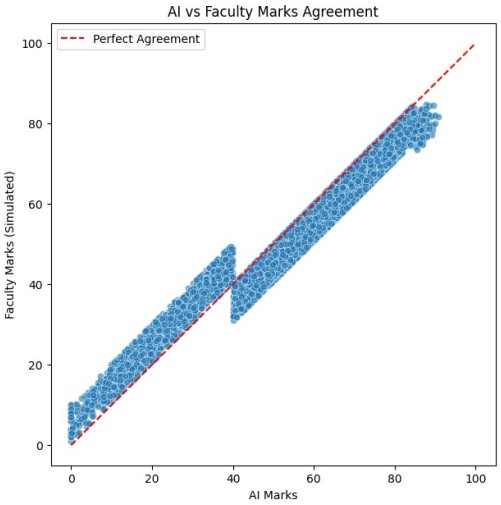

Figure 6: AI vs. Faculty marks agreement. Points align closely with the perfect agreement line.

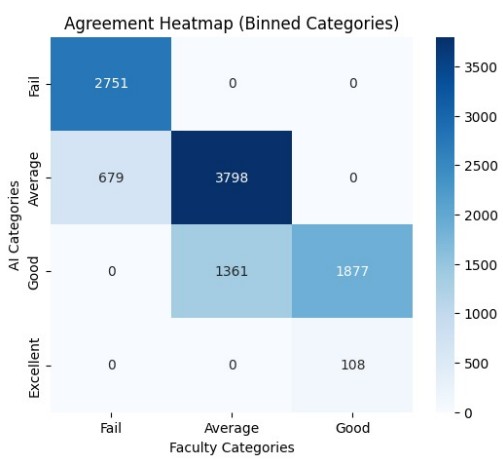

Figure 7: Confusion matrix comparing AI and faculty categories. Stricter faculty grading reduced Excellent cases.

## 5.5 IMPACT OF RETRIEVAL ON GRADING PERFORMANCE

Beyond dataset validation, we also evaluated the role of retrieval augmentation. Figure 8 shows correlation with human scores as the number of evaluated questions increases.

- LLM only: Performs reasonably at first but degrades as more questions are added, showing instability without retrieval support.
- LLM + RAG1: Achieves higher and more stable correlation by leveraging faculty-prepared knowledge bases.
- LLM + RAG1 + RAG2: Provides the highest and most consistent performance, demonstrating the benefit of incorporating supplementary materials with cache-based retrieval.

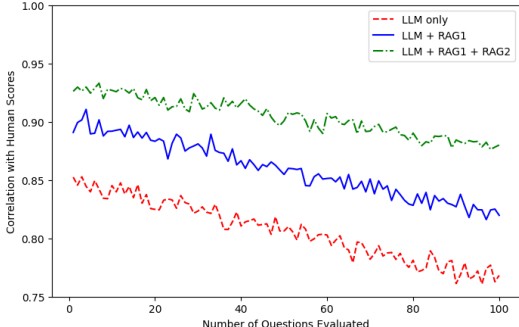

Figure 8: Impact of retrieval on grading performance. Incorporating RAG1 and RAG2 significantly improves stability and correlation with human grading.

# 6 CONCLUSION AND FUTURE WORK

This paper introduced TrueGradeAI, a retrieval-augmented and bias-aware framework for digital examinations that preserves handwritten authenticity while ensuring transparent, auditable, and explainable evaluation. By replacing paper-based workflows with stylus-enabled input, transformer-based handwriting recognition, and retrieval-grounded grading, the system addresses persistent challenges in assessment, including environmental impact, grading latency, and evaluator bias. In comparison with existing platforms, TrueGradeAI distinguishes itself by offering explicit rationales, bias-mitigation strategies, and audit trails, all of which remain limited in current deployments.

Looking ahead, several promising directions emerge. First, the cache mechanism used in the framework can be extended into a reusable knowledge base for students, where frequently retrieved or high-value facts serve as study references beyond examination contexts. Such a resource could transform assessment data into a learning tool, reinforcing student preparation. Second, monitoring students' responses across topics offers an opportunity to automatically identify weaker areas, allowing the system to support teachers in providing targeted interventions. Third, incorporating multimodal handwriting features (stroke dynamics, temporal cues, and visual signals) may further improve recognition robustness across diverse scripts. Finally, expanding retrieval sources to include personalized learning records and conducting field trials across multiple institutions will be essential for validating fairness, scalability, and long-term adoption.

By advancing explainability, accountability, and adaptability in automated assessment, TrueGradeAI repositions examinations as not only a process of evaluation but also a catalyst for continuous learning and equitable educational support.

## REPRODUCIBILITY STATEMENT

We have made extensive efforts to ensure reproducibility of our results. A complete description of the dataset of 10,000 NCERT Class 12 QA pairs and its preprocessing is provided in Section 5, with evaluation metrics (Pearson correlation, Spearman correlation, Cohen's Kappa, and confusion matrix analysis). Model components such as TrOCR for transcription, Gemini-embedding-001 for vector encoding, dual RAG modules with HOT/COLD caches, and Gemini 2.5 Pro for evaluation are detailed in Section 4, along with hyperparameters and similarity thresholds. Runtime statistics and experimental results are reported in Section 5. To facilitate independent verification, anonymized code stubs, preprocessing scripts, and plotting utilities will be made available in the supplementary material. All experiments were run on NVIDIA A100 GPUs (40GB) with 32GB RAM; preprocessing required ∼6 hours, RAG evaluation ∼1.5 hours, and average inference ∼1.2 seconds per QA. Together, these details and materials should enable reliable reproduction and extension of our findings.

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
