# OpenReview forum: "TrueGradeAI: Retrieval-Augmented and Bias-Resistant AI for Transparent and Explainable Digital Assessments"
_ICLR.cc/2026/Conference — ICLR 2026 Conference Desk Rejected Submission_

### Official Review · Reviewer_ZNVQ · 2025-10-30

**Soundness:** 2
**Presentation:** 1
**Contribution:** 2
**Rating:** 2
**Confidence:** 4

**Summary:**

This paper presents an AI-assisted framework for automated grading of student assignments through a retrieval-augmented evaluation pipeline. The framework integrates secure data capture, retrieval-augmented reasoning, and explainable grading, enabling secure, transparent, and interpretable assessment of student responses. To validate its effectiveness, this paper constructs a dataset of 10,000 question–answer pairs from NCERT Class 12 History, Political Science, and Geography materials. Experimental results demonstrate a high correlation between AI-based scores and human faculty evaluations.

**Strengths:**

This paper proposes an AI-assisted automated pipeline for evaluating student assignments, which accelerates feedback cycles and promotes paperless assessment. It further reveals that retrieving textbook knowledge can enhance the accuracy of evaluation.

**Weaknesses:**

While this paper presents a pipeline for AI-assisted grading, it exhibits several notable limitations:

**Lack of literature review.**
This paper includes only around 20 references and provides little discussion of existing work on AI-based assessment or LLM-as-judge systems.

**Limited novelty.**
This paper does not introduce any fundamentally new method or concept. Its core idea—incorporating reference answers and textbook materials through retrieval—is a common and basic practice already adopted by many prior works in automated grading and retrieval-augmented evaluation. The proposed approach lacks clear methodological innovation or differentiation from existing research.

**Insufficient experimental comparison.**
This paper lacks quantitative or qualitative comparisons against existing automated grading systems.

**Narrow evaluation scope.**
The experiments cover only three subjects, excluding key disciplines such as mathematics and linguistics.

**Bias in LLM-as-judge evaluation.**
The framework relies on LLM-based scoring, but it does not discuss potential biases inherent in LLMs. For example, students could craft answers that exploit model preferences or perform prompt injections to obtain “high scores” rather than genuinely correct answers. For example, students could craft answers that exploit model preferences or perform prompt injections to obtain “high scores” rather than genuinely correct answers.

**Questions:**

Minors: There is a citation error in line 311. The figures are not vector graphics. They become blurry and unclear when zoomed in.

Have you tried using other models? Since Gemini-2.5-Pro is quite costly, it may not scale up well.

---

### Official Review · Reviewer_nggL · 2025-10-31

**Soundness:** 1
**Presentation:** 1
**Contribution:** 1
**Rating:** 0
**Confidence:** 4

**Summary:**

This paper describes TrueGradeAI an automated evaluation platform that uses hadn’t writing recognition and a RAG-based retrieval mechanism to automatically grade student exams. The motivation for this is to address the challenges of large scale paper-based exams (paper waste, delay in results sharing, etc). The system is described an an evaluation is presented comparing outputs to manual grading.

**Strengths:**

The main strengths of the paper are:

-- The paper addresses a valid an interesting problem (reducing the inefficiencies of paper-based exams, while maintaining the value of hand-written in-person exams).
-- The platform described is comprehensive.
-- The platform is based on state of the art technologies (e.g. TrOCR and Gemini LLMs).
-- The results of the evaluation presented suggests some promise in automated evaluation as the manual evaluation aligns with AI evaluation.

**Weaknesses:**

The main weaknesses are:

-- The tone of the paper is more like a product fact sheet than an academic paper. For example, Table 1 is very selective is its comparisons (RAG is a technique rather than a feature so why is it sure important?) Similarly much of the paper is a defence of the author's approach rather than a careful evaluation.
-- The language in the paper needs careful review and revision.
-- The evaluation presented is very limited.
-- The inclusion of the content of Figure 5 is unusual the LLM Reasoning and Score text contains spans on nonsensical text and strange characters which suggests the system has challenges.
-- Score generation seems quite simplistic - "A similarity score determines alignment with faculty-prepared answers. If the threshold (e.g., 20%) is exceeded, the answer is marked correct "  - and heavily reliant on a manually set threshold.
-- The evaluation presented is very limited and does not evaluate much of the pipeline described in the paper.

**Questions:**

Could more detail on the evaluation be provided - for example what was the RAG database populated with?

How might OCR errors impact on the ability of the system to mark student work?

It seems that the system has only capacity to return binary evaluations ("If the threshold (e.g., 20%) is exceeded, the answer is marked correct"). how could this be expanded to a full grade range?

**Details Of Ethics Concerns:**

The use of automated ablation systems raises some significant ethical questions that should be addressed in the paper.

---

### Official Review · Reviewer_vZJk · 2025-11-01

**Soundness:** 1
**Presentation:** 2
**Contribution:** 1
**Rating:** 2
**Confidence:** 4

**Summary:**

The paper proposes TrueGradeAI, a digital examination framework that captures handwritten responses, transcribes them with TrOCR, and grades them using a retrieval-augmented LLM pipeline referencing faculty and external materials. It claims to deliver transparent, bias-resistant, and explainable grading.

**Strengths:**

1. The paper tackles highly relevant challenges in education such as evaluator bias, grading inefficiency, and the lack of transparency in digital assessments, positioning the work as socially and practically meaningful within academic evaluation systems.
2. The framework generates fact-linked rationales that enhance interpretability and trust.

**Weaknesses:**

1.  The work lacks genuine technical innovation. It mainly combines existing technologies—TrOCR for handwriting recognition, Gemini embeddings for vector representation, and retrieval-augmented generation with an LLM for scoring—into a single pipeline. None of these components are modified, optimized, or extended to improve accuracy, efficiency, or robustness. The paper makes no attempt to address known weaknesses such as OCR errors, retrieval drift, or LLM hallucinations. As a result, the contribution is purely system integrative rather than novel.
2. The paper provides only a superficial description of the dataset, which is a major limitation. The dataset of 10,000 QA pairs is drawn entirely from NCERT textbooks, but the paper omits details about data creation. Without such information, it is difficult to assess dataset validity, quality, or reproducibility. Moreover, the dataset’s narrow focus on factual, curriculum-aligned content means it cannot adequately evaluate open-ended reasoning or creative responses. This restricts the generalizability of the system’s results and weakens its empirical foundation.
3. The claims of bias reduction and transparency are unsubstantiated. Although the framework includes bias-aware design elements, no experiments are conducted to measure or quantify bias before and after applying these methods. The system’s supposed fairness improvements are therefore theoretical, not demonstrated through data or controlled studies.
 4. The evaluation lacks depth. The authors report correlation metrics against human grading but do not provide ablation studies, baselines, or comparisons with other grading systems (such as GPT-based scorers). Furthermore, the paper does not evaluate the quality or interpretability of the generated rationales through human judgment, which is essential to substantiate claims of explainability.
5. The practical performance and scalability of the system remain unclear. While the paper claims grading latency is reduced from weeks to hours, it does not include runtime benchmarks, cost estimates, or deployment data. There is also no validation from real-world institutional use, making it uncertain whether the system can operate effectively at scale or under exam conditions.

**Questions:**

NA

---

### Official Review · Reviewer_tsHK · 2025-11-01

**Soundness:** 2
**Presentation:** 2
**Contribution:** 2
**Rating:** 4
**Confidence:** 3

**Summary:**

This paper presents TrueGradeAI, a comprehensive, end-to-end framework for digital examinations. The system is motivated by the clear and significant drawbacks of traditional paper-based assessments: environmental waste, logistical complexity, grading delays, and evaluator bias.

The proposed system (Fig 1, 4) is a multi-stage pipeline. It begins with students providing handwritten answers on secure tablets. These are transcribed via a Transformer-based OCR (TrOCR). The core contribution is the evaluation engine: a retrieval-augmented generation (RAG) pipeline that uses a dual-cache system (RAG1 for faculty-provided answers, RAG2 for supporting textbooks) to ground an LLM (Gemini 2.5). The LLM's role is not just to assign a score, but to provide an explicit, evidence-linked rationale for its decision (Fig 5). The final step is a human-in-the-loop review and audit trail, intended to ensure fairness and mitigate bias.

The paper's strengths lie in its excellent system design, which thoughtfully integrates modern ML components to solve a high-impact, real-world problem. The evaluation, which includes a newly created 10,000-pair dataset and a strong quantitative analysis of the RAG component, is also a high point.

However, the paper suffers from two critical, interconnected flaws. First, there is a major disconnect between the system proposed (an end-to-end handwriting-to-grade pipeline) and the system evaluated (a text-to-grade pipeline that skips the noisy OCR step). Second, the paper makes strong claims about being "bias-resistant" but provides no empirical evidence to substantiate this; it mistakes high agreement with human graders for a lack of bias. Given the ICLR venue, the novelty of the core ML components (which are assembled from existing SOTA models) versus the application is also a point of concern.

**Strengths:**

S1. The paper tackles a problem of significant practical and social value.

S2. The evaluation in Section 5.5 (Fig 8) is interesting. It quantifies the value of retrieval, clearly demonstrating that the LLM-only model suffers from performance degradation ("drift"), while the RAG-augmented models are significantly more stable and accurate.

S3. The creation of a 10,000 question-answer (QA) pair dataset (Sec 5.1) is a valuable contribution to the community, even if it is currently limited to factual domains.

**Weaknesses:**

W1. Disconnect Between Proposed System and Evaluation: The paper's entire premise is an end-to-end system that preserves handwriting (Sec 1.2, Fig 1, Sec 3.1). It explicitly names TrOCR (a handwriting OCR model) as a key component (Sec 2.2, 4.2). However, the entire empirical evaluation (Sec 5) is run on a dataset of clean, typed text from NCERT textbooks. The noisy, error-prone, and critical OCR step is completely skipped. The 0.982 Pearson correlation (Fig 6) is for grading perfect text, which says nothing about the performance of the actual end-to-end system when faced with real, messy, and imperfectly transcribed student handwriting. This is a major gap that invalidates the evaluation of the system as-proposed.

W2. Unsubstantiated Claims of Bias Mitigation: The paper's title and abstract prominently claim it is "bias-resistant" and addresses "evaluator bias" (Sec 1.1, 1.3). However, the paper provides zero evidence for this claim. The evaluation (Sec 5.4) measures agreement (correlation, Kappa) with human graders, which is not at all the same as measuring bias. The AI could be perfectly replicating the exact same biases as the human graders. To validate this claim, a different study would be needed (e.g., showing the AI gives the same score to an answer when demographic primers are added, while humans do not). This claim is central to the paper's motivation but is entirely unsupported.

W3. The paper assembles existing, SOTA components: TrOCR, Google's embedding model, and Gemini 2.5 Pro. The dual-RAG cache is a good architecture, but it's not a new algorithm. The authors need to be more precise about what the core machine learning contribution is, distinct from the (very strong) application.

**Questions:**

Q1. Why was the evaluation not conducted end-to-end? The performance of the OCR component is critical. Can you provide any data on the system's accuracy when processing actual handwritten inputs, and how OCR errors cascade to the final grade?

Q2. Can you please provide direct evidence for the "bias-resistant" claim? How was this validated? Simply agreeing with human graders does not prove a lack of bias, as the graders themselves may be biased.

Q3. Could you explicitly state the core machine learning or algorithmic novelty of this work, beyond the (impressive) assembly of existing SOTA models into a new application?

Q4. How do you see this RAG-based framework adapting to more subjective domains like essay writing, where there is no single "correct" fact to retrieve from a textbook?

---

### Official Review · Reviewer_h4qy · 2025-11-03

**Soundness:** 1
**Presentation:** 1
**Contribution:** 1
**Rating:** 2
**Confidence:** 4

**Summary:**

The authors present TrueGradeAI, a paperless exam system that captures stylus handwriting, transcribes it into text, and does grading via an LLM augmented with relevant course material (ie, from answer keys, text books, etc). This grading is done in a three phase process using course material, where the question-answer pair only goes to the next phase if marked incorrect. They claim their framework constitutes a ‘complete digital assessment’ pipeline, provides ‘explainable and reliable scoring’, and has a ‘bias-aware and auditable design’.

The authors describe the features of TrueGradeAI, including a two‑portal product (student/teacher) with biometrics, secure storage, anonymized distribution, and an appeals workflow. Their evaluation centers on a curated 10k NCERT QA dataset with high AI–faculty agreement. The authors further claim retrieval improves correlation versus LLM‑only.

**Strengths:**

There are definitely some things to like about this paper. The domain knowledge is clear in terms of what friction points a student or teacher using this system might encounter (explainability and auditability of the final scores). The issue of the resource drain, not just in material but also working hours for grading, distribution, etc, of exams makes this a very clear issue worth investigating. I also appreciate the checking of general grading agreement via the NCERT QA Dataset to show some evidence that the answers accepted by the system are also the same that the teachers themselves would accept.

**Weaknesses:**

While there are some positives, I’m finding some core issues with the paper that motivate my overall score.

Primarily, it feels like there’s two papers stuffed into one and neither are really given enough detail. In fact, I would even argue that one of those papers that are jammed into the whole just also doesnt really fit at ICLR. For me, the first half of the paper involving the more logistical aspects, ie the student/teacher portal, the face ID, and handwriting recognition feels out of place. My understanding was that neither of the latter were evaluated as well, with the primary evaluation coming from a textbook with the expectation of typed answers.

- I think the agreement numbers look a bit inflated by the setup. For Table 3, wouldn’t there be a significant overlap with any faculty answers already being stored?
- Feels like the thresholds and logic arent justified super well? My impression is that the 3-tier grading system would result in excessive skewing towards positive scores
- While the model for the embedding is named (Gemini) I dont think I saw the actual model used for the system?
- Unless I missed it, the authors don’t actually evaluate the bias-resilience of their system? The unspoken assumption appears to be that automation inherently ‘solves’ this but if anything, the high instructor agreement would imply the opposite.
- It feels like this system would be working with very Personally-Identifiable-Information but there’s no mention really of how this is kept secure
- Just generally related to the first point: no comparison to strong short‑answer/essay graders on public datasets; no breakdowns by question type, answer length, or subject; no qualitative failure cases from the grading pipeline
- I'll also note some roughness with the editing and grammar (ie, frame- work and eval- uator in the abstract)

**Questions:**

- Table 3: Did you find this agreement generalized? Was this with full scores or was partial credit involved? (ie, a student could score .5/1).
- For the three tiered grading, did a single 'correct' override all of the other tiers declaring the answer wrong?
- What model is being used to generate the scoring of the students' answers?
- How is grading bias (or a lack thereof) being measured?
- Figure 8: How is the correlation being calculated here?
- It seems like this system works with alot of sensitive information: How is this stored and kept safe?

---

### Note · Program_Chairs · 2026-01-17
**Submission Desk Rejected by Program Chairs**

The following references in this submission do not refer to real documents and/or have major errors in bibliographic information:

 S. Bailey and D. Meurers. Automated essay grading using machine learning. In Proceedings of the 22nd International Conference on Computational Linguistics, pp. 1-7, 2008.

Avraham Ben-David. Measuring inter-rater agreement for categorical items. Journal of Quantitative Analysis in Education, 5(2):1-20, 2009.

R. Sharma and A. Patil. Assessment inefficiencies: A survey on examination paper wastage. Journal of Educational Resources, 2017.